# Evaluation of an Immunization Protocol Using Bovine Alphaherpesvirus 1 gE-Deleted Marker Vaccines against Bubaline Alphaherpesvirus 1 in Water Buffaloes

**DOI:** 10.3390/vaccines11050891

**Published:** 2023-04-24

**Authors:** Alessandra Martucciello, Anna Balestrieri, Cecilia Righi, Giovanna Cappelli, Eleonora Scoccia, Carlo Grassi, Sergio Brandi, Elisabetta Rossi, Giorgio Galiero, Damiano Gioia, Giovanna Fusco, Francesco Feliziani, Esterina De Carlo, Stefano Petrini

**Affiliations:** 1National Reference Centre for Hygiene and Technology of Breeding and Buffalo Production, Istituto Zooprofilattico Sperimentale del Mezzogiorno, 84131 Salerno, SA, Italy; alessandra.martucciello@izsmportici.it (A.M.); anna.balestrieri@izsmportici.it (A.B.); giovanna.cappelli@izsmportici.it (G.C.); grassicarlo@libero.it (C.G.); sergio.brandi@izsmportici.it (S.B.); giorgio.galiero@izsmportici.it (G.G.); giovanna.fusco@izsmportici.it (G.F.); esterina.decarlo@izsmportici.it (E.D.C.); 2National Reference Centre for Infectious Bovine Rhinotracheitis (IBR), Istituto Zooprofilattico Sperimentale Umbria-Marche “Togo Rosati”, 06126 Perugia, PG, Italy; c.righi@izsum.it (C.R.); e.scoccia@izsum.it (E.S.); e.rossi@izsum.it (E.R.); f.feliziani@izsum.it (F.F.); 3Azienda Sanitaria Locale Salerno, 84014 Nocera Inferiore, SA, Italy; gioiadamiano@tiscali.it

**Keywords:** Water buffalo, gE-deleted marker vaccines, wt-BoHV-1, wt-BuHV-1, immunization protocol

## Abstract

European regulations on the control of infectious diseases provide measures to control *Bovine alphaherpesvirus 1* (BoHV-1) infection in both cattle and buffalo. Owing to the reported serological cross-reactivity between BoHV-1 and *Bubaline alphaherpesvirus 1* (BuHV-1), we hypothesized a new immunization protocol using BoHV-1 gE-deleted marker vaccines could protect water buffalo against BuHV-1. Five water buffaloes devoid of BoHV-1/BuHV-1-neutralizing antibodies were immunized with two commercial BoHV-1 gE-deleted marker vaccines at 0, 30, 210, and 240 post-vaccination days (PVDs). Five additional water buffaloes were used as controls. At 270 PVD (0 post-challenge days (PCDs), all animals were challenged intranasally with wild-type (wt) BuHV-1. The vaccinated animals produced humoral immunity (HI) as early as PVD 30 whereas, in control animals, antibodies were detected on PCD 10. After challenge infection, HI significantly increased in vaccinated animals compared to that in controls. Real-time PCR for gB revealed viral shedding in vaccinated animals from PCDs 2 to 10. In contrast, positive results were observed from PCDs 2 to 15 in the unvaccinated control group. Although the findings indicated the possible protection capabilities of the tested protocol, these findings did not support its protective roles in water buffaloes against wt-BuHV-1.

## 1. Introduction

Water buffalo (*Bubalus bubalis*) is an invaluable livestock heritage in Italy, with 432,068 heads [1], of which over 70% are located in the Campania Region (295,391 heads). Apart from providing buffalo milk used in several typical Italian products, including “mozzarella di bufala campana”, buffalo farming is an essential source of income because of their contribution to the farming community as draught animals.

To date, it is known that water buffalo can be infected with bovine alphaherpesvirus 1 (BoHV-1) and bubaline alphaherpesvirus 1 (BuHV-1) [2,3]. These two viruses belong to the viral family Herpesviridae [4]. The susceptibility of buffalo to these two viral infections is due to the fact that buffalo and cattle have a high genomic identity (>91%) [5]. However, infection with BoHV-1 only causes virological and serological positivity in water buffalo without clinical symptoms. In contrast, in cattle, it causes clinical respiratory symptomatology known as infectious bovine rhinotracheitis (IBR) or a reproductive form (infectious pustular vulvovaginitis or infectious balanoposthitis) [6,7]. In addition, in buffalo calves, BuHV-1 has been associated with respiratory symptoms, loss of appetite, depression, lethargy, and abortion [8,9,10]. Both viral infections have been detected in several cattle and buffalo herds [11] and are risk factors for cross-infections.

Moreover, as these viruses are antigenically and genetically related, it is difficult to diagnose these two infections during IBR control/eradication programs [12]. Recent studies have explored the potential of a newly developed glycoprotein E (gE)-enzyme-linked immunosorbent assay (ELISA) test to differentiate BuHV-1 from BoHV-1 based on their specific infection status [13,14]. However, no commercial vaccine has been developed to date for the control of BuHV-1. Furthermore, using the single BoHV-1 gE-deleted marker vaccine to control BuHV-1 in water buffaloes has been shown to result in cross-protection against BuHV-1 [3].

Therefore, this study aimed to evaluate an IBR marker immunization protocol to vaccine cattle against BoHV-1 [15,16,17] in buffalo species to protect animals against wild-type (wt) BuHV-1.

## 2. Materials and Methods

### 2.1. Cells and Viruses

Madin–Darby Bovine Kidney (MDBK) cells were cultured in Minimum Essential Medium (MEM; Euroclone, Milan, Italy) supplemented with 10% fetal bovine serum (Euroclone, Milan, Italy) at 37 °C. The cells were provided by the Biobanking of Veterinary Resources (BVR, Brescia, Italy) and identified using the code BS CL63. Two viruses were selected for the present study. The Schönböken strain of BoHV-1 (kindly provided by Prof. Martin Beer, Friedrich-Loeffler Institute, Greifswald, Germany) was used for in vitro tests. The wt-BuHV-1 strain [18]; GenBank accession No. KF679678.1] was used for in vitro challenge infections. The viruses were grown in MDBK cells at a 1.5 × 10^8^ median tissue culture infection dose (TCID_50_/mL) calculated using the Reed and Muench method [19]. All in vitro experiments using BuHV-1 and BoHV-1 viruses were performed in a biosafety level-2 laboratory.

### 2.2. Vaccination and BuHV-1 Challenge Animals

The experiments conducted in this study were approved by the Italian Ministry of Health (Authorization number 859/2017-PR) based on European legislation on the protection of animals used for scientific purposes [20]. For this study, ten water buffaloes were utilized from a buffalo herd in southern Italy (Campania region) by calculating the number of animals constituting each group based on our previous paper [2]. Water buffaloes were divided into two groups (A and B) of five animals each. Group A was vaccinated with four doses of two commercial gE-deleted marker vaccines. The first dose was administered at three months of age (time 0) and after one month (30 post-vaccination days, PVDs). At 10 months of age (210 PVD), a second dose of vaccine was administered, and a booster dose was administered at 11 months of age (240 PVD). The first two doses were injected intranasally (i.n.) with a commercial IBR marker live vaccine, whereas the third and fourth doses were inoculated intramuscularly (i.m.) into the neck muscle with a commercial IBR marker inactivated vaccine. Each dose of the vaccine was injected in a volume of 2 mL. Group B served as the unvaccinated control group. Animals in each group were housed in separate pens in an experimental facility with free access to food and water.

At 12 months of age (270 PVD), all water buffaloes were challenge infected with wt-BuHV-1. Each animal was administered 7 × 10^7^ TCID_50/_mL i.n. The animals were observed for 63 days, and rectal temperatures were recorded daily. During this period, the fever was confirmed when the rectal temperature was greater than 38.2 °C [11]. In addition, adverse reactions were monitored by a veterinary practitioner.

During the entire experimental period [0, 30, 210, and 240 PVDs; 0, 2, 4, 7, 10, 15, 30, and 63 post-challenge days (PCDs)], serum samples were collected from each water buffalo and tested for the presence of BuHV-1 and BoHV-1 antibodies. Nasal swabs were obtained from each water buffalo at 0, 2, 4, 7, 10, 15, 30, and 63 PCDs and used for real-time PCR.

### 2.3. Real-Time PCR Reaction

Viral nucleic acid was extracted from nasal swabs using the commercial QIA Symphony DSP Virus/Pathogen Mini Kit (Qiagen, Hilden, Germany) following the manufacturer’s instructions. The highly conserved target region of the UL27 gene, common to all alphaherpesviruses and encoding glycoprotein B (gB), was amplified by real-time PCR. The protocol used is described in the OIE Manual of Diagnostics Tests and Vaccines [18]. All samples were tested in duplicate. A positive control (IBR Los Angeles strain) and a negative control (reagent grade water) were included in the procedure. Then, an internal amplification control (β-actin) was added. A sample was considered positive when its cycle threshold (Ct) value was ≤45.

### 2.4. Virus Neutralization Tests

The virus neutralization tests (VNT) were performed following the protocol described in the OIE Manual of Diagnostics Tests and Vaccines [21]. Briefly, 50 µL of undiluted serum samples and two-fold dilutions of each serum sample were mixed with 50 µL 100 TCID_50_ virus. Tests were conducted in duplicate using wt-BuHV-1 and wt-BoHV-1 (see Section 2.1.) during separate work sessions. Each mixture was then transferred to three wells of 96-well microtiter plates (Nunc^TM^, 96-Well Microplates Polypropylene, Thermo Scientific, Milan, Italy). The samples were incubated at 37 °C for 24 h; 30,000 MDBK cell cultures suspended in 100 µL MEM were added to each well and incubated for 4 days at 37 °C. After incubation, the plates were examined under an inverted tissue culture microscope (Zeiss Axiovert Vert. A1, Zeiss International, Milan, Italy) to determine cytopathic effects. Neutralization titers are expressed as the highest dilution that inhibited cytopathology.

### 2.5. ELISA Tests

To detect antibodies to glycoprotein B (gB) and glycoprotein E (gE) of BoHV-1, sera from each animal were assayed using two commercial ELISA tests [ID Screen^®^ IBR gB competition and ID Screen^®^ IBR gE competition; both from Innovative Diagnostics (Grables, France)]. The data were analyzed using Microplate Manager Software version 6 (Bio-Rad Laboratories S.r.l., Segrate, Italy), and all the results were interpreted per the respective manufacturer’s instruction.

### 2.6. Statistical Analysis

Statistical analysis was conducted for each animal group at all sampling times. Mean neutralizing antibody titers were expressed on a logarithmic base-10 log scale. Shapiro–Wilk test and Wilcoxon–Mann–Whitney non-parametric tests were used to evaluate statistically significant differences. The following were compared: (1) neutralizing antibodies (NA) to BuHV-1 between the group of vaccinated buffaloes (group A) and the group of unvaccinated animals (group B); (2) NA to BoHV-1 between the group of vaccinated buffaloes (group A) and the group of unvaccinated animals (group B). The significance level was set at *p* < 0.05.

## 3. Results

### 3.1. Clinical Response

No clinical signs or adverse reactions were observed in immunized animals during the entire vaccination period. After the challenge, no lesions were detected in the immunized animals. In contrast, in the unvaccinated controls, four animals showed nasal mucus discharge and lesions of the nasal mucosa, consisting of hyperemic mucosa on PCD 2. These clinical signs were observed for five days. In addition, during the entire experimental period, the rectal temperatures in both groups were similar and within the normal ranges.

### 3.2. Virological Investigations

During the challenge infection, gB positivity was detected by real-time PCR, which was evident in group A from PCDs 2 to 7. In contrast, gB positivity was detected by real-time PCR and was evident in group B, from PCDs 2 to 10. Furthermore, the Ct values ranged from 23.27 to 35.05 in the immunized animals during the challenge infection, whereas they varied from 20.08 to 34.11 in the unvaccinated controls (Table 1).

### 3.3. Serological Investigations

An increase in the BuHV-1 NAs titer was detected during the vaccination period. The immunized animals presented a mean titer of 0.90 log_10_ (*p* = 0.0079) on PVD 30. This titer increased to 1.63 log_10_ (*p* = 0.0079) on PVD 240. In addition, the vaccinated animals presented BoHV-1 neutralizing antibodies with a mean titer of 1.63 log_10_ (*p* = 0.0079) on PVD 30, which was further increased to 1.75 log_10_ (*p* = 0.0079) on PVD 240. No neutralizing antibodies against BuHV-1 or BoHV-1 were detected in the control group. On PVD 30, vaccinated animals were seropositive for gB-ELISA and negative for gE-ELISA. These values were maintained up to PVD 240. Seroconversion was not detected in unvaccinated controls (Table 2).

The NA titer against BuHV-1 of the vaccinated animals increased after challenge infection, reaching 3.01 log_10_ (*p* = 0.0027) and 3.25 log_10_ (*p* = 0.0336) on PCD 10 and PCD 63, respectively. The same NA titers were observed against BoHV-1 on PCD 10 and 63. In the control group, NAs against BuHV-1 and BoHV-1 were detected on PCD 15, with an average titer of 1.75 log_10_ and 1.74 log_10_, respectively. These titers increased by 0.99 log_10_ on PCD 63. In contrast, a positive gE signal was detected on PCD 30, which persisted until PCD 63. Furthermore, a positive gB signal was detected throughout the infection challenge period in vaccinated animals. In contrast, in the unvaccinated control animals, antibodies against gB were detected on PCD 10 afterward (Table 3).

## 4. Discussion

Communitarian regulations, such as European Legislation (EU) 2016/429 (“Animal Health Law”) [22] and EU 2018/1629 [23], have been issued to ensure that every member state is officially entirely or partly IBR-free. The amendments of these regulations allow every member state to submit and obtain approval to validate their IBR control programs for the entire territory (or part of it) and provide additional guarantees for bovine trading in their territory. Based on Regulation (EU) 2018/1882, IBR/IPV is listed under the C, D, and E categories, and *Bubalus* ssp. is a sensible species. In addition, the strategy that differentiates vaccinated animals from infected ones (aka DIVA) has been applied to control and eradicate highly prevalent IBR infections [24,25,26,27].

Based on the serological cross-reactivity of BoHV-1 and BuHV-1 [12,28], we hypothesized that using an IBR marker immunization protocol to immunize cattle against BoHV-1 [15], we could induce protection in water buffaloes after wt-BuHV-1 infection. To test this hypothesis, for the first time, in this study, we evaluated a new immunization protocol using two commercial BoHV-1 gE-deleted marker vaccines (live and inactivated) against BuHV-1 in water buffaloes.

The new immunization protocol applied in water buffalo involved the revaccination of animals at 6-month intervals using live and inactivated IBR marker vaccines. This protocol applied in cattle against BoHV-1 showed efficacy in reducing the incidence of gE seroconversion in dairy cattle and, consequently, the herd prevalence of gE- positive animals [15]. In addition, the immunization protocol provided evidence for protection against BoHV-1 challenge infection with a strong immune response [15]. Additionally, the efficacy of live or inactivated IBR marker vaccines in cattle was shown previously [29]. Furthermore, the efficacy of the immunization protocol in cattle was demonstrated in Slovakia’s IBR control program. In particular, Mandelik et al. described an increase in IBR-free farms and a decrease in small (less than ten animals) non-IBR-free herds recorded from 2010 to 2020 [17].

The vaccines used in this study were applied notwithstanding current regulations, as they are only registered for cattle species by the European Commission.

The dosage used in this immunization protocol is the same as that used in cattle (2 mL/vaccine/head). Using this inoculum volume in cattle, fever or adverse reactions can be induced in 1 out of 10,000 vaccinated cattle. Indeed, the findings demonstrated that the investigated vaccines did not induce clinical signs or adverse reactions. The data obtained in this study agree with those in previously reported studies for cattle and water buffaloes [2,3,30,31,32,33], suggesting no risk of adverse reactions following the administration of the vaccines using the proposed protocol. On the contrary, a study by Baccili et al. demonstrated localized pain and heat at the injection site in different experimental groups inoculated with inactivated and thermosensitive BoHV-1 vaccines [34]. Furuoka et al. demonstrated neuroparalysis and death within 7–10 days of routine IBR vaccination [35]. In addition, clinical protection was incomplete in calves vaccinated with inactivated or subunit vaccines, and viral shedding by the vaccinated calves continued for different days after challenge infection [36]. Allcock et al. reported outbreaks of clinically infectious bovine rhinotracheitis in two dairy herds routinely vaccinated with a live IBR marker vaccine (i.m.) approximately 6–8 weeks before the outbreak [37].

In both groups, rectal temperatures remained at normal physiological values after the challenge infection. Furthermore, no animals showed any clinical signs of disease during the entire experimental period. Nasal swabs showed viral shedding in both groups after PCD 2. The post-vaccination and post-challenge rectal temperatures recorded in this study were similar to those reported in other studies [2,3,38]. However, the clinical results of the control group observed in 4/5 animals (nasal mucus discharge and hyperemic mucosa) differed from those obtained by Scicluna et al., where no clinical signs were observed [38]. However, similar results were observed in two other studies. In particular, Petrini et al., reported nasal mucus discharge, pseudomembranes, dyspnoea and caught in 3/5 animals, while Montagnaro et al. described sero-mucous nasal secretion in 5/5 animals [3,4].

Furthermore, concordant with that of a previous study [4], the viral excretion in this study was detected up to PCDs 10–15 in both groups. Nevertheless, these results differ from those demonstrated in our previous study [3], where the vaccinated animals did not excrete the virus in contrast to the control group that released the virus up to PCD 7. These results are probably because the animals infected in this study were younger (12 months) than those infected by Petrini et al., 2021 (17 months) [3]. This could have affected the host’s immune response of the host.

gB-ELISA and NAs (BuHV-1-BoHV-1) were detected for the first time on PVD 30. The same antibodies were observed in the vaccinated group until the end of the experiment compared with the control group. Studies using modified live or inactivated gE-deleted marker vaccines have reported similar results [2,3]. Furthermore, the serological results obtained in this study agree with those in previous studies on cattle immunized with gE-deleted marker vaccines [30]. Several studies have reported increased levels of antibodies in buffalo calves following immunization with inactivated vaccines against bovine ephemeral fever [39], foot-and-mouth disease [40], hemorrhagic septicemia caused by *Pasteurella multocida* [41], and hemorrhagic septicemia–mastitis caused by *Staphylococcus aureus* and *Streptococcus agalactiae* [42]. In contrast, Lemaire et al. [43] reported a low NA induction against BoHV-1 after vaccination with inactivated gE-deleted marker vaccines.

The immunized water buffaloes showed negative gE-ELISA results during the vaccination period. This result showed that neither BuHV-1 nor BoHV-1 field viruses circulated among the animals during vaccination. However, due to challenge infection with BuHV-1, all animals seroconverted to gE PCD 30. In the present study, we detected gE-ELISA positivity in vaccinated water buffaloes on PCD 30, whereas seroconversion to gE-ELISA was observed in the control group on PCD 10. The results obtained in this study conducted on water buffalo were similar to those obtained in cattle. The studies involved the use of a gE-deletion marker vaccine. Furthermore, seroconversion of gE 2–4 weeks after the experimental infection has been reported in several papers [39,44,45,46]. In contrast, Montagnaro et al. [3] reported no seroconversion of buffaloes after challenge infection. This could be attributed to the shorter post-challenge evaluation duration (15 days), during which the animals may not have had enough time to obtain positive.

Taken together, the experimental evaluation of the tested immunization protocol with gE-deleted marker vaccines against BuHV-1 in water buffaloes does not support the hypothesis that the tested gE-deleted marker vaccines induce protection in water buffaloes after challenge infection with wt-BuHV-1. In other words, the findings of this study demonstrated that the tested immunization protocol did not protect the animals from wt-BuHV-1 challenge infection, as the vaccinated animals shed the virus up to PCD 7. However, the presence of NAs after the vaccination period and the subsequent increase in NAs after challenge infection are indicators of possible protection. The real evidence of protective immunity would be the reduction of challenge virus replication, which should result from the sum of the humoral and cellular immune responses.

The present study has several limitations. An increase in the γδ and αβ CD4^+^ T lymphocytes, CD21^+^ B lymphocytes, and CD4^+^/CD8^+^ ratio was evident in water buffaloes after challenge infection with BoHV-1 on PCDs 7 and 10 [2]. However, in this study, we did not investigate the cell-mediated immune response activated after the challenge with wt-BuHV-1. Furthermore, a smaller number of animals were used in this study to comply with the current European regulations on animal use in clinical trials and based on the 3R principle. Nevertheless, the number of animals used in this study was sufficient to statistically analyze the results.

Overall, these findings suggest further studies to improve the proposed immunization protocol. In particular, subsequent studies are required to evaluate other: (i) IBR-deleted marker vaccines following the wt-BuHV-1 challenge; (ii) immunization schemes; (iii) vaccine doses (3×, 4×).

## 5. Conclusions

The study’s findings indicate that the IBR marker immunization protocol used to vaccinate cattle against BoHV-1 does not protect water buffalo against wt-BuHV-1 challenge infection. In this study, we used an inoculum volume of vaccines normally used for cattle. However, this volume did not protect the water buffalo against the challenge infection, indicating that this volume was too low to protect the animals from the challenge infection. These findings suggest that further studies are required to improve the protocol used herein by increasing the vaccine dose (3× or 4×), using other marker vaccines and modifying the vaccine schedule used.

## Figures and Tables

**Table 1 vaccines-11-00891-t001:** Results of BuHV-1 detected using gB real-time PCR of viral DNA extracted from nasal swabs in animals immunized against BoHV-1 using IBR gE-deleted marker vaccines and challenge infected with wt-BuHV-1 strain.

Group	Post-Challenge Day (PCD)							
	0 *	2	4	7	10	15	30	63
A	−	23.27 ^a,b^	28	35.05	−	−	−	−
	−	25.88	27.86	34	−	−	−	−
	−	24.57	25.01	34.3	−	−	−	−
	−	24.71	23.74	−	−	−	−	−
	−	26.6	26.98	33.15	−	−	−	−
B	−	27.22	25.89	32.26	−	−	−	−
	−	23.69	22.46	31.84	−	−	−	−
	−	20.08	23.48	28.86	29.14	−	−	−
	−	24.14	22.97	31.92	−	−	−	−
	−	27.79	19.95	28.69	34.11	−	−	−

* PCD 0 corresponds to PVD 270; −, negative result; ^a^ cycle threshold (Ct) values of ≤45 were considered positive; ^b^ OIE Manual of Diagnostic Tests and Vaccines for Terrestrial Animals. Group A, vaccinated water buffaloes; Group B, unvaccinated control animals.

**Table 2 vaccines-11-00891-t002:** Antibody response of water buffaloes immunized against BoHV-1 using gE-deleted marker vaccines.

Group	Test	Post-Vaccination Day (PVD)			
		0	30	210	240
A	gE-ELISA	−	−	−	−
	gB-ELISA	−	+	+	+
	NA ^a^	<1.00	0.90	1.63	1.63
	NA ^b^	<1.00	1.63	1.75	1.75
B	gE-ELISA	−	−	−	−
	gB-ELISA	−	−	−	−
	NA ^a^	<1.00	<1.00	<1.00	<1.00
	NA ^b^	<1.00	<1.00	<1.00	<1.00
	*p*-value ^c^		0.0079	0.0079	0.0079
	*p*-value ^d^	−	0.0079	0.0079	0.0079

^a^ NA, neutralizing antibody titer (mean value) to BuHV-1; ^b^ NA, neutralizing antibody titer (mean value) to BoHV-1; ^c^
*p*-value < 0.05 indicates the significant differences in NA to BuHV-1 titer between vaccinated (group A) and unvaccinated (group B) water buffaloes; ^d^ *p*-value < 0.05 indicates the significant differences in NA to BoHV-1 titer between groups A and B.

**Table 3 vaccines-11-00891-t003:** Antibody response of water buffaloes immunized against BoHV-1 using IBR gE-deleted marker vaccines and challenge infected with wt-BuHV-1 strain.

Group	Test	Post-Challenge Day (PCD)							
		0 *	2	4	7	10	15	30	63
A	gE-ELISA	−	−	−	−	−	−	+	+
	gB-ELISA	+	+	+	+	+	+	+	+
	NA ^a^	1.70	1.72	1.70	2.38	3.01	3.01	3.01	3.25
	NA ^b^	1.87	1.87	1.99	2.29	3.01	3.01	3.01	3.25
B	gE-ELISA	−	−	−	−	−	−	+	+
	gB-ELISA	−	−	−	−	+	+	+	+
	NA ^a^	<1.00	<1.00	<1.00	<1.00	<1.00	1.75	1.72	2.74
	NA ^b^	<1.00	<1.00	<1.00	<1.00	<1.00	1.74	1.71	2.73
	*p*-value ^c^	0.0052	0.0050	0.0050	0.0050	0.0027	0.0053	0.0052	0.0336
	*p*-value ^d^	0.0052	0.0052	0.0052	0.0050	0.0027	0.0053	0.0052	0.0071

* PCD 0 corresponds to PVD 270; ^a^ NA, neutralizing antibody titer (mean value) to BuHV-1; ^b^ NA, neutralizing antibody titer (mean value) to BoHV-1; ^c^ *p*-value of < 0.05 indicates the significant differences in NA to BuHV-1 titer between vaccinated (group A) and unvaccinated (group B) water buffaloes; ^d^ *p*-value of <0.05 indicates the significant differences in NA to BoHV-1 titer between groups A and B.

## Data Availability

The data presented in this study are available in this article.

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
