# Peer review of "Evaluation of an Immunization Protocol Using Bovine Alphaherpesvirus 1 gE-Deleted Marker Vaccines against Bubaline Alphaherpesvirus 1 in Water Buffaloes"

_vaccines, 2023, doi:10.3390/vaccines11050891_

Round 1

Reviewer 1 Report

The manuscript describes the antibody response of water buffalos immunized with two gE-deleted BoAHV-1 vaccines and challenged with BuHV-1. The authors evaluated the humoral immune response by seroneutralization and ELISA test and tested viral shedding in nasal secretions by qPCR.

The findings of the study revealed that the immunization protocol did not protect against BuHV-1 infection.

The experimental design is appropriate for the study. Some minor details should be added as indicated below. The manuscript is well-written however some points should be clarified. The main comment is what is the rationale for using the immunization protocol with to different gE-marker vaccines? If there is some pilot study or bibliographic reference on this point it should be added to the introduction. The age of the animals included in the study should also be added.

Some other comments are detailed below:

- NA: the meaning of the abbreviation should be indicated the first time it appears in the text

- In results section. gB positivity. It can be assumed that the authors refer to the result of qPCR. However, it would be more appropriate to indicate that viral nucleic acid was detected or that gB was detected…

- Tables 2 and 3. Table legend should be re-written. Instead of a,b for neutralizing antibodies A and B should be used. Furthermore, it seems unnecessary to indicate p value significance for c and d in the legend.

- Lines 210-211. In the immunized animals, viral excretion was detected up to a PCD of 10. In contrast, virological positivity was detected in the control group up to PCD 15. What do the authors refer to with virological positivity?

- Lines 233-234. Nonetheless, water buffaloes were exposed to BuHV-1 and/or BoHV-1 seroconversion in gE-ELISA. This sentence should be re-written since it cannot be understood what does ¨exposed to BuHV-1 and/or BoHV-1 seroconversion¨ means.

Author Response

Review 1

The manuscript describes the antibody response of water buffalos immunized with two gE-deleted BoAHV-1 vaccines and challenged with BuHV-1. The authors evaluated the humoral immune response by seroneutralization and ELISA test and tested viral shedding in nasal secretions by qPCR. The findings of the study revealed that the immunization protocol did not protect against BuHV-1 infection.The experimental design is appropriate for the study. Some minor details should be added as indicated below. The manuscript is well-written however some points should be clarified.

  • The main comment is what is the rationale for using the immunization protocol with to different gE-marker vaccines?

  • Dear Review, thank you for your suggestions. The rationale of the study is to test an immunization protocol used in cattle (two different BoHV-1 marker vaccines) in the buffalo species against BuHV-1;

If there is some pilot study or bibliographic reference on this point it should be added to the introduction. The age of the animals included in the study should also be added.

  • Dear Review, thank you for your suggestions. Unfortunately, there is no pilot study, and there are no references. Furthermore, the animals were vaccinated for the first time at 3 months of age (see line 88);

 Some other comments are detailed below:

- NA: the meaning of the abbreviation should be indicated the first time it appears in the text

- Dear Review, thank you for your suggestions. We have inserted the definition of NA (Neutralizing antibodies) for the first time in line 142;

- In results section. gB positivity. It can be assumed that the authors refer to the result of qPCR. However, it would be more appropriate to indicate that viral nucleic acid was detected or that gB was detected…

- Dear Review, thank you for your suggestions. We modified the sentence as follows: “During the challenge infection, gB positivity detected by Real-time PCR was evidenced in group A from PCDs 2 to 7. In contrast, gB positivity detected by Real-time PCR was evident in group B, from PCDs 2 to 10”(lines 159-161);

- Tables 2 and 3. Table legend should be re-written. Instead of a,b for neutralizing antibodies A and B should be used. Furthermore, it seems unnecessary to indicate p value significance for c and d in the legend.

- Dear Review, thank you for your suggestions. We have used “a,b” to indicate the neutralizing antibodies obtained from the virus neutralization tests set up with BuHV-1 and BoHV-1, respectively. We do not think of inserting “A,B” in place of “a,b” as this could be confusing with the group identifier “A,B”. Furthermore, we think to leave the letters “c,d” as they indicate the p-value obtained between group A and B indicates the significant differences in NA to BuHV-1/BoHV-1 titer between vaccinated (group A) and unvaccinated (group B) water buffaloes;

- Lines 210-211. In the immunized animals, viral excretion was detected up to a PCD of 10. In contrast, virological positivity was detected in the control group up to PCD 15. What do the authors refer to with virological positivity?

- Dear review, thank you for your suggestions. We have deleted lines 210-211.They are already dictated in line 217;

- Lines 233-234. Nonetheless, water buffaloes were exposed to BuHV-1 and/or BoHV-1 seroconversion in gE-ELISA. This sentence should be re-written since it cannot be understood what does ¨exposed to BuHV-1 and/or BoHV-1 seroconversion¨ means.

- Dear Review, thank you for your suggestions. We have modified the sentence as follows: “However, challenge infection with BuHV-1, all animals seroconverted to gE PCD 30” (lines

Reviewer 2 Report

With the intention to vaccinate water buffaloes against bubaline alpha herpesvirus 1 the presented study is an extension of a previously published paper of the same group. The key difference is the intranasal vaccination at the age of 3 and 4 months and the challenge with a field strain of BuHV-1.

However, the rational of the immunization protocol has not been explained and the induction of local immunity was neither addressed (e.g. IgA determinations in nasal secretions) nor discussed. If the whole study was about the value of immunizing younger animals locally, a group of animals immunized parenterally only was missing. This also in face of the different challenge strain as compared to the previous publication. Between the lines, it can be read that a successful vaccination is expected to result in sterile immunity (no virus shedding after challenge) and not only in reduction of clinical signs. How this could be achieved was not outlined in the study, nor in a rational to choose the presented immunization protocol, nor in the discussion. As such, the manuscript appears as a descriptive presentation of results after chosing 1/X immunization protocols and gives no hints how to optimize or to improve the vaccination of water buffaloes.

Minor points

L33 which experimental conditions do you mean?

L60ff Since the principle that BoHV-1-specific marker vaccines result in cross protection has already been shown by the same group: what was really new in the current study? Was it a new immunization protocol (what was different to Petrini et al. 2021?) or was it the use of different vaccines? What is meant with “optimized” (L63): is it the age of animals when they are vaccinated for the first time, or is the use of intranasally applied MLV vaccines, or is it the heterologous prime/boost approach? … to achieve exactly what? (sterile immunity, reduction of clinical signs, duration of clinical signs, etc….).

L48ff „BoHV-1 only causes virological and serological positivity in water buffalo without clinical symptoms.“ This is in contrast to your previously published paper (Petrini et al. 2021) where a BoHV-1 challenge induced clinical symptoms. Please comment on this

L52ff „BuHV-1 has been associated with respiratory symptoms, loss of appetite, depression, lethargy, and abortion [8-10].“ This was not the case in the experimentally challenged animals. Is that due to the age of the challenged animals?

L60 delete „sing“ and include BoHV-1 between marker and vaccines

L214ff: please discuss this finding

L217: these results are not discussed: Was it due to the different age of the animals or the new immunization protocol?

L233ff I don’t understand the sentence: „Nonetheless, water buffaloes were exposed to BuHV-1 and/or BoHV-1 seroconversion in gE-ELISA.“ Please rephrase.

L261ff how could this proposed improvement look like?

Author Response

Review 2

With the intention to vaccinate water buffaloes against bubaline alpha herpesvirus 1 the presented study is an extension of a previously published paper of the same group. The key difference is the intranasal vaccination at the age of 3 and 4 months and the challenge with a field strain of BuHV-1.

However, the rational of the immunization protocol has not been explained and the induction of local immunity was neither addressed (e.g. IgA determinations in nasal secretions) nor discussed. If the whole study was about the value of immunizing younger animals locally, a group of animals immunized parenterally only was missing. This also in face of the different challenge strain as compared to the previous publication.

  • Dear Review, thank you for your suggestions. The rationale of the study is to test an immunization protocol used in cattle (two different BoHV-1marker vaccines) in the buffalo species against BuHV-1. In addition, the detection of IgA in nasal secretions was not included in the experimental protocol.

Between the lines, it can be read that a successful vaccination is expected to result in sterile immunity (no virus shedding after challenge) and not only in reduction of clinical signs. How this could be achieved was not outlined in the study, nor in a rational to choose the presented immunization protocol, nor in the discussion. As such, the manuscript appears as a descriptive presentation of results after chosing 1/X immunization protocols and gives no hints how to optimize or to improve the vaccination of water buffaloes.

  • Dear Review, thank you for your suggestions. The rationale of the study is to test an immunization protocol normally used in cattle (two different BoHV-1 marker vaccines) in the buffalo species against BuHV-1. In general, if an immunization protocol is effective against a viral agent, the animals should show no clinical symptoms after the challenge infection and should not shed virus. They should produce protective humoral immunity. In our study, since the above is a general evaluation of vaccination protocols, it was not included in the manuscript. However, we have reported the results obtained and discussed them with the results of other manuscripts. Regarding the improvement and optimization of the protocol in the discussion the following sentence was reported:
  • "Overall, these findings suggest further studies to improve the proposed immunization protocol. In particular, subsequent studies are required to evaluate other: (i) IBR-deleted marker vaccines following the wt-BuHV-1 challenge; (ii) immunization schemes; (iii) vaccine doses (3x or 4x). (Lines 298-301).

 Minor points

L33 which experimental conditions do you mean?

  • Dear Review, thank you for your suggestions. We have modified the sentence as follows:

 Although the findings indicated the possible protection capabilities of the tested protocol, these findings did not support its protective roles in water buffaloes against wt-BuHV-1”. (Lines 32-34).

L60 ff Since the principle that BoHV-1-specific marker vaccines result in cross protection has already been shown by the same group: what was really new in the current study? Was it a new immunization protocol (what was different to Petrini et al. 2021?) or was it the use of different vaccines?

  • Dear Review, thank you for your suggestions. The rationale of the study is to test an immunization protocol used in cattle (two different BoHV-1 marker vaccines) in the buffalo species against BuHV-1. In contrast, in the study conducted by Petrini et al., 2021 the aim was the safety and efficacy of an inactivated marker vaccine against BoHV-1 in water buffalo. This study differs from Petrini et al. 2021 which evaluates not a single vaccine but an immunizing BoHV-1 protocol routinely used in cattle and applied in the water buffalo against BuHV-1. The protocol involved using two different BoHV-1 marker vaccines, one live and the other inactivated, both administered to water buffaloes with the first dose inoculated at three months of age and the last at 11 months of age.  

What is meant with “optimized” (L63): is it the age of animals when they are vaccinated for the first time, or is the use of intranasally applied MLV vaccines, or is it the heterologous prime/boost approach? … to achieve exactly what? (sterile immunity, reduction of clinical signs, duration of clinical signs, etc….).

  • Dear Review, thank you for your suggestions. By optimized, we mean that as no vaccine against BuHV-1 was available, we optimized a BoHV-1 marker vaccine protocol used in cattle against BoHV-1 to protect buffaloes against BuHV-1.

L48ff „BoHV-1 only causes virological and serological positivity in water buffalo without clinical symptoms.“ This is in contrast to your previously published paper (Petrini et al. 2021) where a BoHV-1 challenge induced clinical symptoms. Please comment on this

  • Dear Review, thank you for your suggestions. Petrini et al., 2021 describe that BoHV-1 in the buffalo species in adults does not cause clinical symptoms. For a better understanding, we report an extract from the manuscript by Petrini et al., 2021.

  • “The BoHV-1 infection that causes severe losses to the cattle industry worldwide is associated with two different clinical syndromes, namely infectious bovine rhinotracheitis (IBR) and infectious pustular vulvovaginitis (IPV). In addition, it is also associated with a variety of clinical signs, including fever, dyspnea, conjunctivitis, nasal discharge, vaginitis, balanoposthitis, abortions, enteritis, and encephalitis [3,4]. In contrast, in water buffalo adults, although the virological and serological positivity of BoHV-1 has been demonstrated, the clinical signs of disease have not been reported [5].”

L52ff „BuHV-1 has been associated with respiratory symptoms, loss of appetite, depression, lethargy, and abortion [8-10].“ This was not the case in the experimentally challenged animals. Is that due to the age of the challenged animals?

  • Dear Review, thank you for your suggestions. We do not know why clinical symptoms attributable to BuHV-1 only occur in buffalo calves (not in adults). We can speculate that in buffalo calves, the immune system is not well developed and, therefore, the animals are more susceptible to infection.

L60 delete „sing“ and include BoHV-1 between marker and vaccines

Dear Review, thank you for your suggestions. We deleted the term “sing” and added the “BoHV-1” word (Line 62).

L214ff: please discuss this finding

Dear Review, thank you for your suggestions. We discussed the clinical results obtained in the control group. In particular, we have inserted the following sentence:

“However, the clinical results of the control group observed in 4/5 animals (nasal mucus discharge and hyperemic mucosa) differed from those obtained by Scicluna et al., where no clinical signs were observed [34]. However, similar results were observed in two other studies. In particular, Petrini et al. reported nasal mucus discharge, pseudomembranes, dyspnoea and caught in 3/5 animals, while Montagnaro et al. described sero-mucous nasal secretion in 5/5 animals [3,4].” (Lines 226-231).

L217: these results are not discussed: Was it due to the different age of the animals or the new immunization protocol?

Dear Review, thank you for your suggestions. We inserted the following sentence:

“These results are probably due to the animals infected in this study were younger (12 months) than those infected by Petrini et al., 2021 (17 months) [3]. This could have affected the host's immune response.”(Lines 235-238).

L233ff I don’t understand the sentence: „Nonetheless, water buffaloes were exposed to BuHV-1 and/or BoHV-1 seroconversion in gE-ELISA.“ Please rephrase.

- Dear Review, thank you for your suggestions. We have modified the sentence as follows:

“However, challenge infection with BuHV-1, all animals seroconverted to gE PCD 30.” (lines 270-271);

L261ff how could this proposed improvement look like?

- Dear Review, thank you for your suggestions. We think that the immunization protocol could be improved by changing the type of marker vaccines, trying other immunization schemes and testing different vaccine doses (3x or 4x).

Reviewer 3 Report

The authors present a very interesting study about immunization of water buffaloes in Italy.  Almost the entire world likes mozzarella di bufala campana. Here are a few comments for improvement.

1.  Materials and Methods.  Section 2.2. 

Add the commercial names of the live and the killed vaccines and give the name of the pharmaceutical company.  Also give the recommended dosage of each of the 2 vaccines, when administered to a cow.

2.  Results, Section 3.1.  

State whether the dosage of each of the 2 vaccines was the same dosage as recommended for a cow. What percent of cattle have a low-grade fever after receiving either of the 2 vaccines?  Were the authors surprised that none of the immunized buffaloes had a low-grade fever? Provide comment in the text.  

3.  Discussion, lines 172-178.

            Please state here again whether the dosage of each of the 2 vaccines was the same dosage as recommended for cattle?  If the same dosage was used in water buffalo, that may offer an explanation as to why the immunization protocol was not effective in water buffalo.  Because of the species barriers, a vaccine dosage of perhaps 3 to 4x higher for each of the 4 vaccinations in water buffalo may have been worth testing.  Please explain the dosages used and why those dosages were selected.

4.  Conclusion, line 251.

            See comment 4.  Mention in Conclusion what dosage regimen was used.  In particular, was the regimen the same as used in cattle? Does the fact that none of the buffaloes had a fever post-immunization suggest that the vaccine dosage was too low?

Author Response

Review 3

The authors present a very interesting study about immunization of water buffaloes in Italy.  Almost the entire world likes mozzarella di bufala campana. Here are a few comments for improvement.

  1. Materials and Methods.  Section 2.2. 

Add the commercial names of the live and the killed vaccines and give the name of the pharmaceutical company.  Also give the recommended dosage of each of the 2 vaccines, when administered to a cow.

- Dear Review, thank you for your suggestions. Unfortunately, we prefer not to include commercial names in the manuscript. This is due to that the research did not involve the pharmaceutical company that produces the vaccines, and for this reason, as we are not authorized to disclose the trade name, we cannot include the data you requested. Furthermore, the dosage that should be administered to cattle is the same as that used for water buffaloes (Each dose of the vaccine was injected in a volume of 2 mL.) 

  1. Results, Section 3.1.  

State whether the dosage of each of the 2 vaccines was the same dosage as recommended for a cow. What percent of cattle have a low-grade fever after receiving either of the 2 vaccines?  Were the authors surprised that none of the immunized buffaloes had a low-grade fever? Provide comment in the text.  

- Dear Review, thank you for your suggestions. The dosage used in this immunisation protocol is the same as that used in cattle (2 ml/vaccine/head). With this inoculum volume in cattle, fever or adverse reactions can be induced in 1 out of 10.000 vaccinated cattle. This sentence has been added in lines 207-209.

  1. Discussion, lines 172-178.

            Please state here again whether the dosage of each of the 2 vaccines was the same dosage as recommended for cattle?  If the same dosage was used in water buffalo, that may offer an explanation as to why the immunization protocol was not effective in water buffalo.  Because of the species barriers, a vaccine dosage of perhaps 3 to 4x higher for each of the 4 vaccinations in water buffalo may have been worth testing.  Please explain the dosages used and why those dosages were selected.

- Dear Review, thank you for your suggestions. We have started to evaluate the same doses used in cattle to immunize buffalo, as we had no scientific data available on this immunization protocol. Furthermore, we agreed with you to try further doses by increasing the inoculum volume to assess a better protective response.

  1. Conclusion, line 251.

            See comment 4.  Mention in Conclusion what dosage regimen was used.  In particular, was the regimen the same as used in cattle? Does the fact that none of the buffaloes had a fever post-immunization suggest that the vaccine dosage was too low?

- Dear Review, thank you for your suggestions. We have modified/integrated the conclusion section as follows:

“In this study, we used an inoculum volume of vaccines normally used for cattle. However, this volume did not protect the water buffalo against the challenge infection, indicating that this volume was too low to protect the animals from the challenge infection. These findings suggest that further studies are required to improve the protocol used herein by increasing the vaccine dose (3x or 4x), using other marker vaccines and modifying the vaccine schedule used.”(Lines 305-310). 

Round 2

Reviewer 2 Report

none

Author Response

Dear Review, thank you for your suggestions. We edited the article by Editage Cactus (premium editing).